# A high Q piezoelectric resonator as a portable VLF transmitter

Mark A. Kemp [1], Matt Franzi[1], Andy Haase[1], Erik Jongewaard[1], Matthew T. Whittaker[2], Michael Kirkpatrick[3] & Robert Sparr[3]

Very low frequency communication systems (3 kHz–30 kHz) enable applications not feasible at higher frequencies. However, the highest radiation efficiency antennas require size at the scale of the wavelength (here, >1 km), making portable transmitters extremely challenging. Facilitating transmitters at the 10 cm scale, we demonstrate an ultra-low loss lithium niobate piezoelectric electric dipole driven at acoustic resonance that radiates with greater than 300x higher efficiency compared to the previous state of the art at a comparable electrical size. A piezoelectric radiating element eliminates the need for large impedance matching networks as it self-resonates at the acoustic wavelength. Temporal modulation of this resonance demonstrates a device bandwidth greater than 83x beyond the conventional Bode-Fano limit, thus increasing the transmitter bitrate while still minimizing losses. These results will open new applications for portable, electrically small antennas.

[1] SLAC National Accelerator Laboratory, 2575 Sand Hill Rd., Menlo Park, CA 94025, USA. [2] Gooch and Housego, LLC., 676 Alpha Drive., Highland Heights, OH 44143, USA. [3] SRI International, 333 Ravenswood Avenue, Menlo Park, CA 94025-3493, USA. Correspondence and requests for materials should be addressed to M.A.K. (email: mkemp@slac.stanford.edu)

The prevalence of human portable or autonomous vehicle platforms has significantly increased the demand for small, efficient transmitters[1]. Particularly attractive, very low frequency (VLF) signals attenuate <6 dB/1000 km within the Earth-ionosphere waveguide and can penetrate tens of meters into seawater or dirt. Conventional transmitter techniques are inadequate due to large size and high loss. We show that a strain-based, piezoelectric transmitter can overcome many of the fundamental limitations of conventional electrically small antennas (ESA). These transmitters can resonate in a very small footprint while exhibiting low losses[2,3].

Traditionally, a disadvantage of passive high-Q antennas was low bandwidth. Utilizing piezoelectricity as the radiating element allows us to dynamically shift the transmitter resonant frequency. Therefore, high total Q (low loss) no longer constrains the system bandwidth. These are our fundamental advancements: achieving an exceptionally high system Q with no external impedance matching network and an effective fractional bandwidth beyond the passive Bode-Fano limit[4]. Although demonstrated at VLF, this concept straightforwardly scales to other frequency bands.

A significant challenge for ESAs with a wavenumber-length product much <1 is a high radiation Q, $Q_A$, which considerably limits the radiation efficiency[5,6]. In a lossy antenna, the total Q is $Q_t = 1/(1/Q_m + 1/Q_A)$ where $Q_m$ encompasses all non-radiation losses within the antenna system. If $Q_m$ is much less than $Q_A$, then $Q_t$ is approximately equal to $Q_m$ and therefore, to maximize the transmitter efficiency, $\eta = Q_m/Q_A$, $Q_m$ must be maximized. Assuming a wavenumber-length product of $7.5 \times 10^{-5}$, the minimum $Q_A$[7–9] is between $3 \times 10^{12}$ and $3 \times 10^{13}$ (see methods). Therefore, to have a measurable increase of efficiency, $Q_m$ must be very large.

Bulky impedance matching networks compound antenna inefficiency. Consider an electric dipole antenna made up of a copper wire normal to a ground plane. To impedance match to a 10-cm long, 10 mA $m_{rms}$, 35.5 kHz antenna, a 10.5 H, 125 kV inductor is needed. The size and loss of this matching network, even assuming a Q of 1000, greatly exceed the antenna itself, making this technique non-viable. Active non-Foster matching networks, while shown to improve system bandwidth, are unwieldy at high voltages[10–12]. A potential solution is acoustically resonant transmitters[3,13]. Radiation has been measured from vibrating quartz resonators[14–16] and piezo-magnetic or multiferroic antennas have both improved ESAs[2,3,17]. An advantage of strain-based antennas is acoustic resonance in a device with physical dimensions much less than the electromagnetic wavelength, potentially removing the need for large, external impedance-matching elements.

Passive, high Q transmitters have small fractional bandwidths,

$$B = \frac{1}{Q_t}\left(\frac{\text{vswr} - 1}{\sqrt{\text{vswr}}}\right), \qquad (1)$$

as they are bound by the Bode-Fano limit[4]. Here, the voltage standing wave ratio, vswr, is assumed to have a value of 2. A common metric for evaluating ESAs is the bandwidth-efficiency product[18,19],

$$B\eta = \frac{1}{Q_t}\left(\frac{\text{vswr} - 1}{\sqrt{\text{vswr}}}\right)\left(\frac{Q_t}{Q_{A,\min}}\right) \qquad (2)$$

For a passive antenna, this relationship simplifies to $B\eta = 1/(\sqrt{2}Q_{A,\min})$ which is a function of $Q_{A,\min}$, itself a function of the antenna electrical size. This implies that electrically small, low-loss transmitters are also low bandwidth and therefore, by the Shannon–Hartley theorem, have limited data bitrates[20].

Direct antenna modulation (DAM) decouples bandwidth from $Q_t$[21–27]. In one embodiment, the resonant frequency is actively shifted coincident with changes in the input drive frequency. DAM enables operation at a frequency outside the fractional bandwidth of the passive antenna. In a frequency shift keying (FSK) modulation scheme, the carrier and hop frequencies each correspond to a different resonant frequency which changes at the FSK rate. Because an active transmitter is not a linear time-invariant system, the Bode-Fano limit does not constrain the bandwidth. If both the frequency separation, δf, and $Q_m$ are maximized, a larger bandwidth-efficiency product is possible,

$$B_\eta = \left(\frac{\delta f}{f_c}\right)\left(\frac{Q_m}{Q_A}\right) \qquad (3)$$

As well as high bandwidth and efficiency, the radiated signal magnitude should be maximized. The electric dipole moment in a bulk piezoelectric resonator scales as $\sim dT$ where $d$ is the piezoelectric charge constant and $T$ is stress. The effective $d$, allowable stress, and $Q_m$ should be maximized. The lumped $Q_m$ of the resonator system includes mounting losses, external dampening, and internal losses in the piezoelectric material itself[28]. The Y∠36° cut of single crystal lithium niobate (LN) is advantageous due to a yield stress >50 MPa as well as low intrinsic losses in bulk length-extensional modes, the second mode of which couples minimally to the mounting points. In this mode, thermoelastic dissipation and Akhiezer damping are low. Figure 1 shows the electric dipole moment and external electric fields of this vibration mode. The LN is supported only at two points near the longitudinal center and an input signal applied across the metalized end of the crystal and an adjacent coaxial toroid couples power into the resonator. The radially coupled fields excite the length-extensional mode. At resonance, the input impedance is primarily resistive and both the velocity at the end of the crystal and the directly proportional dipole moment are maximal.

This manuscript highlights a conceptual demonstration of an active piezoelectric transmitter. Both experiments and simulations illustrate how the efficiency of the transmitter system can be increased while not constraining the bandwidth. In addition, a significant magnetic field is measured in the near field and it drops off consistent with an electric dipole.

## Results

To demonstrate this concept, experiments are performed with a 9.4-cm-long, 1.6-cm-diameter LN crystal mounted within a vacuum chamber while monitoring the crystal end velocity. The velocity measurement is non-intrusive to the resonator operation and can be used to calculate the dipole moment. Multiphysics simulations show a linear correlation of velocity with dipole moment near resonance. We use this attribute to more easily characterize transmitter behavior in a controlled laboratory setting.

A measured $Q_t$ of ~300,000 is found from an exponential fit to the velocity ring-down (see Fig. 2). Assuming this $Q_t$, a multiphysics simulation[29] calculates the dipole moment, surface and near fields, and induced stress. For a peak dipole moment of 14.1 mA-m (including ground image currents), the simulated average Von Mises stress is 60 MPa, voltage across the crystal is 125 kV, endplate velocity is 3 m s$^{-1}$, input impedance is 110 Ω, average power dissipation is 120 mW, and the input voltage is 5.3 V. The multi-physics simulation is compared to measured data (see Fig. 3). The simulation calculates both the input impedance magnitude and the peak output velocity which closely correspond to the measured values. This particular data set is in the very bottom of the low frequency range, but is representative of <30 kHz VLF simulations and data separately obtained. It is anticipated that this concept scales at least an order of magnitude below and above in carrier frequency.

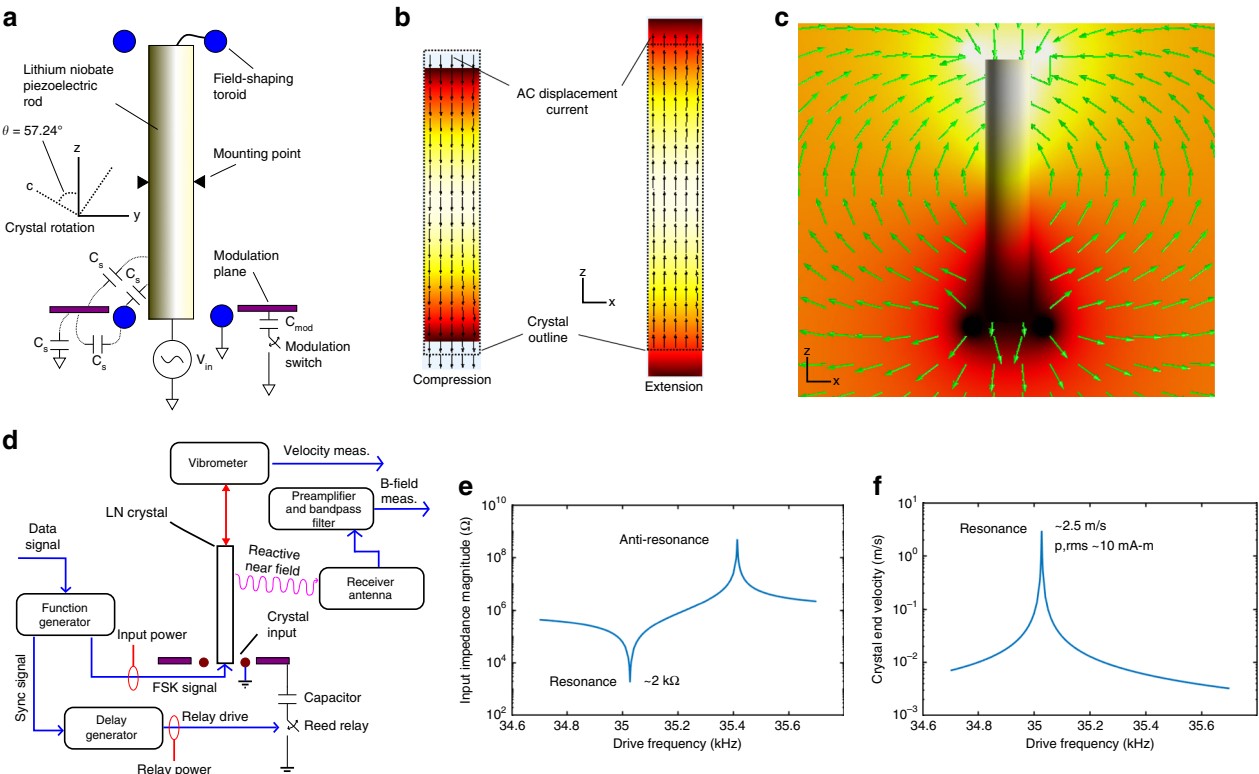

**Fig. 1** Illustration of how a piezoelectric resonator can be used as a transmitter. **a** Schematic of the lithium niobate rod mounting and excitation mechanisms, **b** Mechanical displacement magnitude (in color, magnitude exaggerated for clarity) and the induced electric displacement vectors (arrows), **c** Electric potential magnitude in color along with electric field vectors, **d** Electrical schematic of input and output measurements, **e** Simulated input impedance magnitude versus frequency, **f** Representative simulations of velocity and electric dipole moments versus frequency

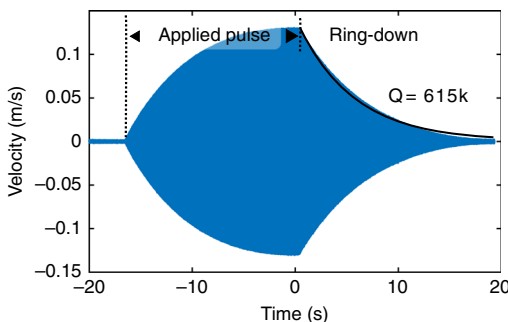

**Fig. 2** Measured ring-down waveforms with the modulation electrical relay closed. Waveform taken in a vacuum background at 150 mV peak applied. With the electrical relay open, the measured Q is about 300k

The piezoelectric resonator is a harmonic oscillator with parameters such as stiffness, mass, and external capacitance determining the resonant frequency[30]. This dependence of resonant frequency on stray capacitance enables the DAM technique. A time-varying capacitance can modulate the resonant frequency outside the passive system bandwidth. A conductive "modulation plate" capacitively couples to the piezoelectric device and ground. A discrete capacitor couples the modulation plate to an electrical relay which shorts and opens this capacitance to ground coincident with the change in the input signal frequency. The two input frequencies match the resonant frequency either with the relay open or closed. The modulation mechanism must not spoil the Q. Efficient modulation promptly converts the energy resonating at one frequency to the second frequency. Velocity or electric dipole magnitude during modulation should be approximately the same as when the transmitter resonates at only one frequency. If one tone is substantially higher or lower magnitude than the other during modulation, then the tuning of the input drive frequency to the two resonant frequencies is not matched. Also, the average input power to the crystal ideally is constant regardless of the modulation rate.

The input signal frequency is swept with the electrical relay first in the closed position, then in the open position (see Fig. 4). Differing relay losses result in a different Q (600k versus 300k) for the two states. To achieve approximately constant amplitude during modulation, the higher-Q signal is driven slightly off resonance. A 50% duty cycle FSK waveform is input with frequency transitions synchronized to the electrical relay opening or closing. For the "without DAM" case, the relay remains closed. Figure 5 shows velocity with an FSK rate of 0.05 Hz both with and without DAM. Without DAM, the crystal slowly charges and discharges depending upon the drive frequency, while the amplitude with DAM is relatively constant. Only one frequency is distinguished in the without DAM case and both frequencies are clear with DAM. As the FSK rate is increased, the amplitude with DAM remains high and both frequencies are easily distinguished (see Fig. 6 and supplementary Fig. 3). The passive fractional bandwidth of a system with a $Q_t$ of 300,000 is $2.4 \times 10^{-6}$, or 84 mHz at a carrier frequency of 35.5 kHz. These results demonstrate a 7 Hz modulation depth, which is >83 times the passive Bode-Fano limit.

With DAM, the input impedance magnitude and input average power do not significantly vary versus FSK rate (see Fig. 7). Only at the lowest FSK rates are the differences between the relay open versus closed states distinguishable (due to the different Q for

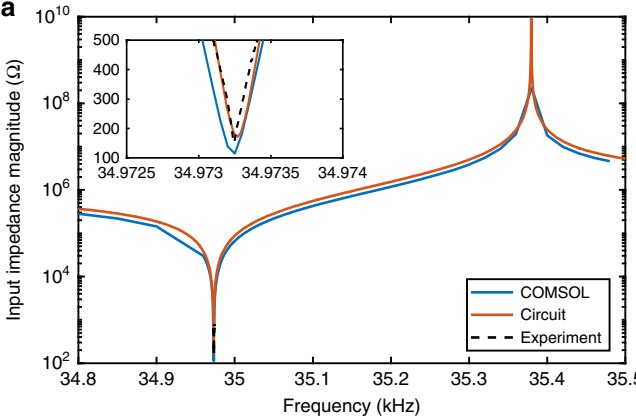

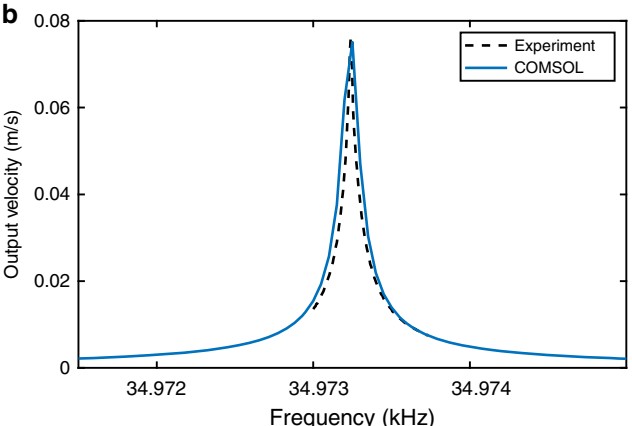

**Fig. 3** Comparison of COMSOL model, equivalent circuit model, and experimental measurements. The experimental data is shifted down in frequency by 594.8 Hz to directly compare to the simulations. **a** The input impedance magnitude is compared for all three data sets, **b** for the COMSOL modal used in **a**, the output velocity is compared to experiment

each state). The upper limit for FSK rate for the present system is likely determined by the relay switching characteristics. Here, the relay switches in about 50–100 μs. Ideally, the relay switching time should be much less than the FSK period. The slight increase in input impedance and decrease in peak velocity magnitude versus FSK rate is attributed to the increasing influence of switching losses.

The robustness of the modulation technique is apparent considering the long relay switching time. One cycle of 35.5 kHz is 28 μs, on the order of the relay switching time. In addition, the turn on jitter is expected to also be 10's of μs. Therefore, the relay switching time is not well synchronized to the drive frequency and one or more RF cycles pass as the relay switches. In fact, both simulations and experiments show that synchronization at <1 ms is not critical for FSK rates up to ~200 Hz (see supplementary Fig. 1). Another characteristic which simplifies implementation, is that the relay does not switch the full resonator voltage or current; only a portion of the full system energy is commutated by the modulation mechanism. Therefore, relatively low-voltage and slow mechanical relays can be effective for DAM.

Electrical breakdown as well as crystal fracture strength limits can bound the attainable dipole moment with a piezoelectric transmitted. As such, crystal defects are minimized, surface electric fields are reduced through field shaping, and the chamber is filled with hexafluoroethane (an electrically insulating gas). For this test, gas damping limits the system $Q_m$ to ~30,000. When energizing to high field, the crystal shatters at ~65 MPa peak, with a calculated electric dipole moment of 6.8 mA-m,rms and

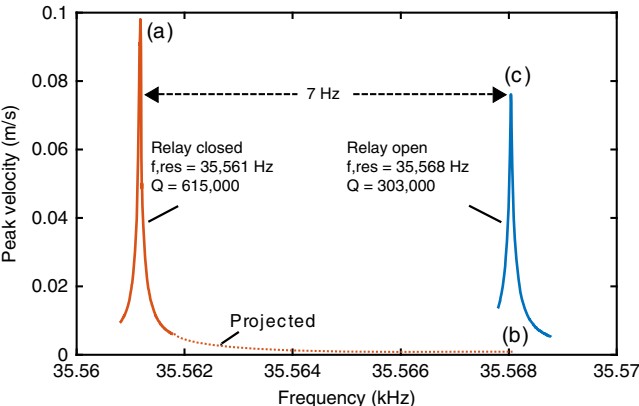

**Fig. 4** Measured peak crystal velocity at two values of external capacitance (the modulation relay open or closed). The passive bandwidth of each individual curve is <0.1 Hz. Without direct antenna modulation, one would operate between points "a" and "b" on the red curve 1. Direct antenna modulation allows operation across both curves, at the points of highest field, "a" and "c."

measured average power dissipation of 0.8 W (see supplementary Fig. 6).

In addition, a portable system open to the ambient environment is used to measure the electric and magnetic field versus range (see supplementary Fig. 7). Input lead lengths and effects from RFI are minimized. The magnetic field drops off as $1/r^2$ and the electric field drops off as $1/r^3$ as anticipated in the electric dipole near field (see Fig. 8). We have separately confirmed that the magnetic field continues to fall as $1/r^2$ to >80 m. A fit to the electric field data gives the measured value of dipole moment, 7.5 mA-m. Discrepancies between the measured and calculated dipole moments occur due to effects of nearby structures, deviations of the resonator from an ideal dipole, and drifts of the Q during operation.

## Discussion
These results illustrate only one embodiment of this technique. Geometry optimization will result in a wider modulation frequency separation. Different carrier frequencies are attainable by varying the length of the piezoelectric element. Further, higher mode excitement will enable operation at higher harmonic frequencies. Piezoelectric arraying is straightforward, particularly due to strong coupling and the ability to phase-lock. The piezoelectric material is not limited to LN and can be tailored to the application. Varying operation in air, vacuum, or other background gasses can help balance between heat removal, high-field operation, and vibration damping. Even though we present a several order of magnitude improvement beyond conventional transmitter techniques, we are far from a limit on the antenna Bη product and the achievable dipole moment.

## Methods
**Lithium niobate fabrication, mounting, and characterization**. The LN crystals are grown with the standard Czochralski process. The congruent composition produces crystals with uniform composition and therefore minimal property variations (see US patent #5,310,448). The rods are cut to the standard longitudinal extension mode Y∠36° orientation and are rough cut to cuboids of 20 mm × 20 mm × 94 mm.

Using a DC sputtering system in a $50 \times 10^{-3}$ Torr argon background, a bonding layer of 10–100 Å thick titanium is applied to each rod face. Prior to venting, an electrical contact and sealing layer of 10–100 Å thick gold is applied. A substrate heater raises the LN to about 400 °C before and during the coating process. The coatings are applied prior to finishing and polishing the crystal OD so masking is not required. After coating, the LN is ground to rough shape using a rotating 180 grit then a 600 grit diamond sintered plate. After, the diameter is ground by hand against glass plates using 15 μm and then 9 μm aluminum oxide. Next, a lathe is used to grind with successively finer grit sizes using wet silicon carbide sand paper.

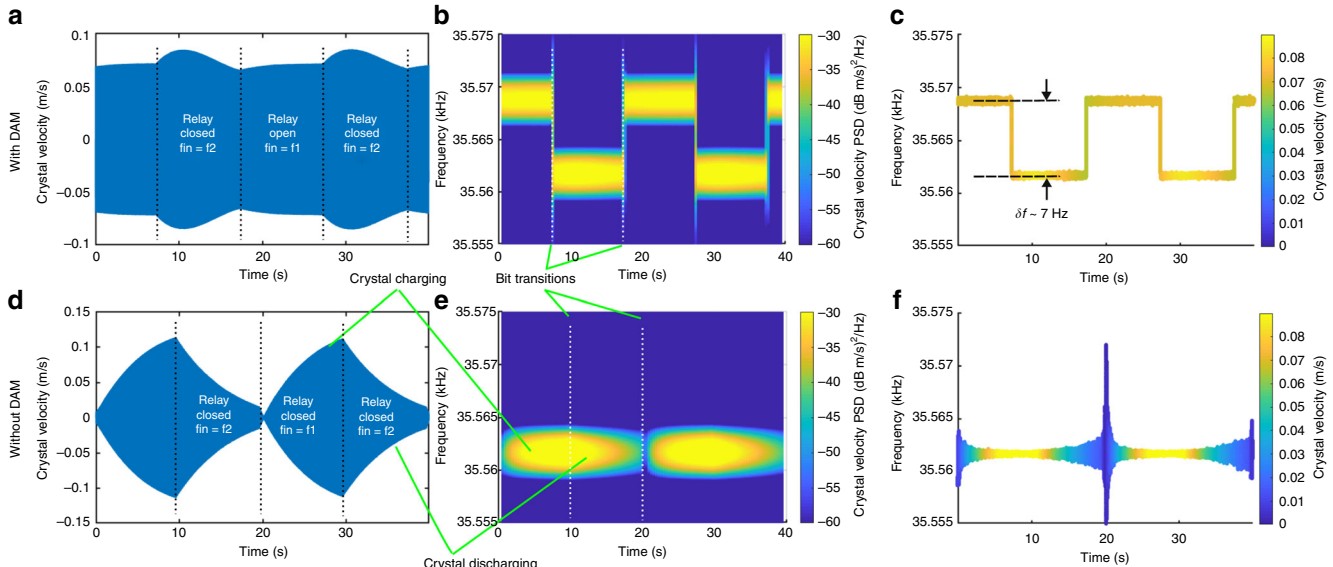

**Fig. 5** Measured effect of direct antenna modulation (DAM) to modulate a >300,000 Q resonator with a frequency separation of 7 Hz and an frequency shift keying rate of 0.05 Hz. **a–c** are with direct antenna modulation (DAM), **d–f** are without DAM. **a, d** are the time-domain measurements, **b, e** are the time-frequency spectrograms, and **c, f** are the Hilbert transforms. Note that one of the resonant frequencies is intentionally detuned to minimize the effect of different Q on the amplitude of the output signal

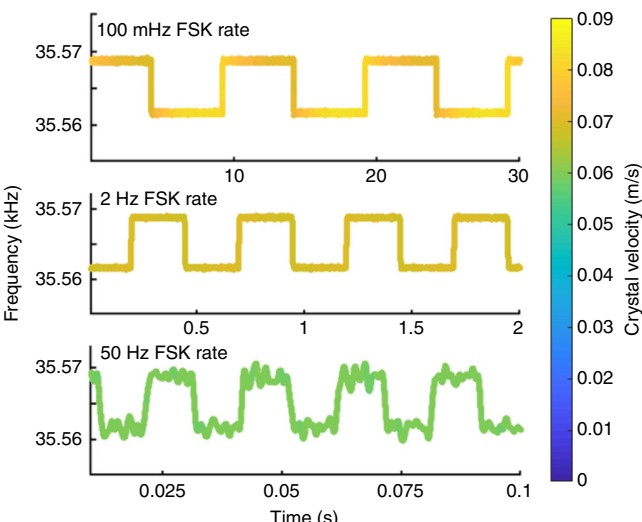

**Fig. 6** Measured crystal velocity for three different frequency shift keying (FSK) rates using direct antenna modulation. The two frequencies are clearly distinguishable for all rates and the velocity magnitude decreases only slightly from about 0.07 m/s peak at 100 mHz to about 0.06 m/s peak at 50 Hz

Finally, a lathe was used to polish the rod using a 1-μm aluminum oxide slurry and a polyurethane pad.

Each metalized end of the LN rod has a 0.003 inch copper wire bonded to the surface via structural epoxy and silver paint. On one end, the thin wire attaches to the input signal. On the other end, the wire attaches to a field shaping toroid. Using this common high-voltage design technique, the toroids are at or near the potential of the LN rod corners and therefore spatially distribute the equipotential lines. This decreases the peak surface electric field on the LN for a given dipole moment, thereby increasing the achievable dipole moment prior to high voltage flashover. The toroids on either end of the LN are mechanically supported by alumina posts and do not contact the LN rod. The LN is suspended by two fused silica rods located at the longitudinal center of the LN rod. The fused silica rods are supported by vertical alumina rods (see supplementary Fig. 4). For tests in the vacuum chamber, the pressure is kept to less than $2 \times 10^{-7}$ Torr.

To measure the crystal velocity, a Polytek OFV-5000 with OFV-552 sensor head laser Doppler vibrometer shines on one end of the LN rod. In laboratory tests, the

input signal is supplied to the crystal via a Tektronics AFG3021C arbitrary function generator. The signal current is monitored by a Pearson current transformer and the voltage is directly monitored by a Lecroy 44MX 400 MHz, >125 kS s$^{-1}$ oscilloscope. Waveforms are post-processed in Matlab. The modulation system is comprised of a metal modulation plate, a mechanical relay (COTO 9913), and a 60 pF capacitor. The relay coils are driven by a SRS DG645 trigger generator which is synchronized to the RF input FSK modulation signal.

**Multiphysics and circuit simulation.** The piezoelectric system is modeled using the FEM multi-physics software COMSOL with the MEMS toolbox[29]. Standard material properties are used, with the isotropic loss tangent of LN set to the effective $1/Q_t$ found from experiment. For a given input voltage and frequency, the frequency domain response is used to calculate parameters such as peak electric fields, stress within the LN, dipole moment, velocity, and input impedance.

To model time domain problems, an equivalent circuit model is used[30]. This lumped element model (see supplementary Fig. 8) consists of four sub-circuits including the driver input, the piezoelectric equivalent circuit, radiative coupling, and modulation circuit. A time-varying voltage source with a 10-Ω output impedance is assumed for the input driver. The equivalent circuit follows a conventional template for piezoelectric elements while radiation impedance is characterized by parallel resistive (radiation resistance) and capacitive (output coupling) elements to ground. The modulation circuit is a capacitive network represented by a single element that is either coupled or isolated from the antenna circuit via an electrical switch. Manufacturer's data was used for the switching time and open and closed contact resistance.

An exact analytical solution is used for the circuit model. The Laplace transform for each of the impedance elements along with the initial conditions is calculated and the circuit loop equations are found. The switch is modeled as a resistor. The inverse Laplace transform is solved with the values of the circuit elements input into the loop equations. The time dependent currents and voltages are then calculated for each element. At the end of an FSK or switch cycle, the final conditions of the circuit were then used as input conditions for subsequent simulations.

This lumped element circuit is tuned by matching the input impedance and output voltage to the frequency domain COMSOL results (see Fig. 3). The circuit solver is then used to model the transient response of the antenna during filling, discharge, and switching. supplementary Figs. 1 and 2 compare the time-dependent output voltage of the circuit model to experimental results both with and without DAM. The element values used in the circuit model are $R_{in} = 10$ Ω, $C_{in} = 3.7$ pF, $C_m = 92.12$ fF, $L_m = 239.6041$H, $R = 85$ Ω, $C_o = 1.4001$ pF.

**Field measurement at range.** For the range measurements, a microprocessor controlled MOSFET H-bridge translates DC voltage from a battery pack into a square wave. This waveform is fed directly to the input of the piezoelectric transmitter. The waveform frequency is controlled via a Bluetooth serial connection. Lead lengths from this power processing unit to the piezoelectric transmitter were minimized (<1–2″ total) to reduce RFI.

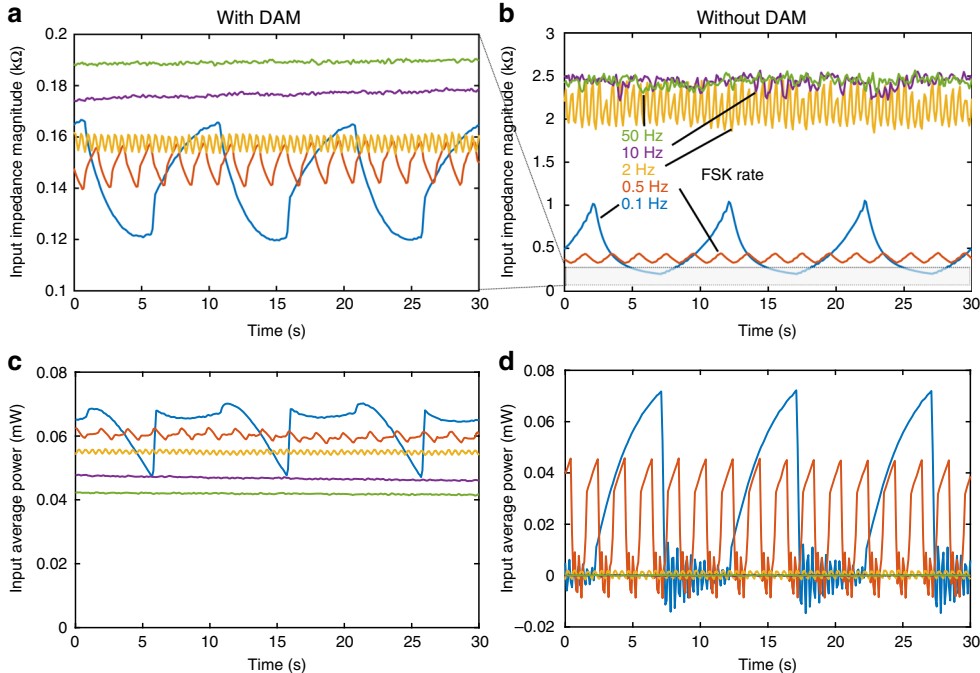

**Fig. 7** Effect of frequency shift keying (FSK) rate on input impedance and power. Measured time varying impedance (**a**, **b**) and input power (**c**, **d**) for without direct antenna modulation (DAM) (**a**, **c**) versus with DAM (**b**, **d**). Various FSK rates are shown. As the FSK rate increases, the input power goes to zero for the without DAM case as the crystal does not build up to full resonance at either frequency. The impedance slightly increases for the DAM case versus increasing FSK rate, and the input average power slightly decreases. However, the change in both is <50%, consistent with the high measured velocity versus FSK rate shown in Fig. 6

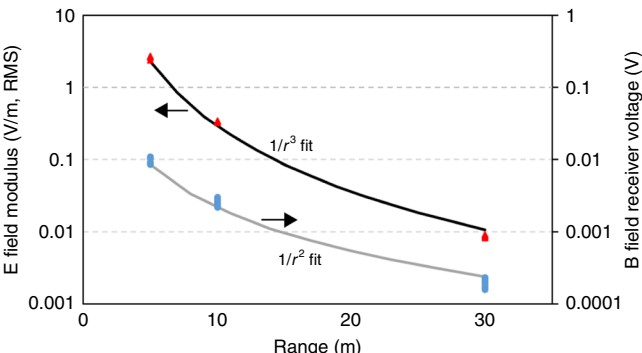

**Fig. 8** Measured E and B field versus range. The data points show each individual measurement and the lines are fits to the data

The electric field is measured using a probe made up of a 2-cm metal stud mated to an SMA female connector. The signal is fed to a Stanford Research Systems preamplifier model SR560. A calibrated transfer function is generated by immersing the probe in the electric field generated by a 1 m × 1 m × 0.09 m parallel plate capacitor. A 3 kHz, 12 dB/octive high-pass filter is used to attenuate RFI primarily from power line harmonics. The signal is fed to a LeCroy WaveJet 354 with a 250 kHz low pass filter and a sampling rate of 250 kS s$^{-1}$.

The magnetic field is measured using a 200 turn, 1.2 m diameter, air-core, 6 cm long solenoid. A grounded aluminum foil (cut at one point to ensure induced current is not shorted out) is placed around the coil to attenuate electric field pickup. The two ends of the coil are differentially fed (with a common ground referenced to the receiver shield) to a Stanford Research Systems preamplifier model SR650. The preamplifier has a 115 dB/octive bandpass filter from 29 kHz to 38 kHz. The stop band attenuation is >80 dB and the gain is set between 40 and 60 dB. The signal is measured with a Tektronix TDS5054B oscilloscope with a sampling rate of 125 kS s$^{-1}$. This magnetic field received is uncalibrated and is included as a relative measurement.

For both the electric field and magnetic field data, the CW signals are measured for 4 s total. At each range data point, four repetitions are measured. Each distance measurement was repeated twice. In addition, at each point the background was measured with the input signal turned off. Data is post processed by taking the DFT of each 4-s long interval. The magnitude of the signal at the frequency of

interest is both found by calculating the RMS value of the DFT within a bandwidth of 10 Hz as well as measuring the peak of the DFT. Both methods yield similar results. The SNR for all measurements is >20 dB.

To provide a baseline signal, a coil transmitter antenna was used to generate a reference magnetic field. This coil was driven at about 35.5 kHz, the approximate resonant frequency of the piezoelectric antenna. Measurements of this coil confirmed that the magnetic field dropped off as $1/r^3$, consistent with a magnetic dipole.

To measure the effect of RFI from the power processing unit, a wirewound resistor with a resistance approximately equal to the input impedance of the piezoelectric transmitter at resonance is attached directly to the output of the power processing unit. The voltage and frequency are tuned to the same values used in the piezoelectric transmitter measurements. The measured voltage on the receiver with just the wirewound resistor is 24 dBV below the signal when measuring just the piezoelectric transmitter (see supplementary Fig. 5). The noise floor for all the magnetic field measurements is <−90 dB V/Hz$^{0.5}$.

**Calculated $Q_A$, antenna efficiency, and bandwidth-efficiency.** For the calculation of $Q_A$ (angular frequency times the average stored energy in the near field divided by the radiated power), the radiation from the piezoelectric element is assumed equivalent to a simple electric dipole wire. This is supported by the similarity of the simulated piezoelectric displacement current to the current in a typical copper antenna. In addition, the measured magnetic field in the near field drops off as $1/r^2$, consistent with an electric dipole.

As we do not measure radiated power at the far field, to compare to other ESAs, we calculate an estimate of $Q_A$ based upon on two different formulations. $Q_{A,min}$ is the theoretical lowest possible $Q_A$ for a given size antenna. McLean[8] shows,

$$Q_{A,min} = \frac{1}{ka} + \frac{1}{(ka)^3},\qquad(4)$$

where $a$ is the effective antenna radius and $k$ is the free space wavenumber. Conventionally, the length $a$ is defined as the radius of the smallest sphere which completely encapsulates the antenna[7,8]. However, proximity to ground and the associated image charges produce a monopole-like antenna with double the effective length[19,31]. For simplicity, we assume a perfectly conducting ground plane with the full antenna length defining the radius of the enclosing sphere. With a $k \times a$ value of $7.5 \times 10^{-5}$, the calculated $Q_{A,min}$ is $3 \times 10^{12}$. This estimate is a lower bound for ESAs as it is derived from evanescent modes in the near field assuming the antenna completely fills the spherical bound. Thiele suggests that due to the inherent super-directivity of ESAs, a more accurate $Q_{A,min}$ derivation for dipole antennas uses the far-field radiation pattern[9]. The calculated $Q_{A,min}$ with this methodology is $3 \times 10^{13}$.

With an assumed $Q_A$ of $3 \times 10^{12}$–$3 \times 10^{13}$, given that the power radiated is much smaller than the electromechanical loss, the radiation efficiency can be calculated

using $\eta = Q_m/Q_A$. Our measured $Q_m$ is in the range of $3 \times 10^5$ and $6 \times 10^5$ putting the calculated radiation efficiency between $2 \times 10^{-7}$ and $1 \times 10^{-8}$. Comparatively, a passive state of the art antenna using a conventional matching inductor (although we claim this is unpractical in 10 cm length scales) would have a maximum $Q_m$ of 1000[31,32]. Provided the passive antenna has a similar $Q_A$, the radiation efficiency would $3 \times 10^{-10}$–$3 \times 10^{-11}$, 300 times lower than the piezoelectric transmitter. We demonstrate a modulation frequency separation using DAM of 7 Hz with a passive system bandwidth of 84 mHz. The use of DAM enhances the bandwidth-efficiency product over the passive case by 7 Hz/84 mHz = 83 times.

## Data availability

The data that support the findings of this study are available from the corresponding author on reasonable request.

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

## Acknowledgements

We thank D. Miller for assistance with measurement setup and assembly, L. Lee for providing diagnostic equipment to the research, and T. Brand for assistance in polishing the crystals used for this research. Additionally, C. MacKay, D. Jundt, L. Gordon, J. Cooper, and A. Bahr are thanked for their contributions to the broader experimental program. This work was supported in part by the US Department of Energy under Contract DE-AC02-76SF00515 and in part by the US Defense Advanced Research Projects Agency under Contract DE-AC02-76SF00515.

## Author contributions

M.A.K. conceived the idea for this experiment. M.A.K. and M.F. simulated the transmitter and performed modulation experiments. A.H. and M.A.K. designed and coordinated the fabrication of the experimental aparati. E.J. and A.H. performed the mechanical and thermal analysis. M.T.W. executed the crystal growth and fabrication. M.K. and R.S. designed and built the power processing unit. M.K., M.A.K., M.F., and R.S. performed range measurements. All authors reviewed and commented on the manuscript.

## Additional information

**Competing interests:** M.A.K., M.F., E.J. and A.H. have filed a patent on the core technology in this manuscript. The remaining authors declare no competing interests.

