## [Peer Review File · Nature Communications]

Reviewers' comments:

Reviewer #1 (Remarks to the Author):

The authors demonstrate an ultra-low loss lithium niobate piezoelectric electric dipole driven at acoustic resonance that radiates with higher efficiency compared to previous devices. Very Low Frequency (VLF) communication systems (3 kHz-50 kHz) are a hot topic in current literature. The paper is well presented and discussed. A nice piece of work. Deserves publication as it is.

Reviewer #2 (Remarks to the Author):

Dear authors,

I have read very carefully your paper about a high-Q piezoelectric resonator as a portable VLF transmitter. I have found very impressive the amount of work spent in designing, fabricating, testing and modelling the proposed device. I think that that your antenna demonstrates the possibility of achieving better performance with electrically-small antennas (ESAs) with respect to the actual state-of-the-art and, consequently, could have future practical applications for VLF transmitters. Furthermore, the analysis of the performance is, in my opinion, carried out correctly and provides enough details for a reproducibility at laboratory level. References are appropriate, too, and embrace the most recent results in the field of ESAs.

However, I have some observations that maybe could help enhance the paper:

1. Lines 41-42: you wrote "Large, external impedance-matching elements are superfluous when the device is self-resonant". I do not agree completely: in principle what you stated is true, as an ESA exhibits a low radiation resistance (sometimes close to zero) and an impedance transformer could be sufficient, but in general impedance matching depends also on the application, for example the self-resonating dipole could be connected to a highly-reactive impedance chip.

2. Lines 75-76: you wrote "The velocity measurement is non-intrusive to the resonator and directly correlates to the dipole moment". I think that this aspect should be explained more in detail.

3. Line 78: you talk about near field: what is the distance used for your measurements (I think something like a couple of meters)? As an electrical/electronic engineer, I think that it could be even

more useful to the scientific community to provide, if possible, some measurements at a far greater distance (not at far field of course, which is not practicable at this frequency).

4. Lines 112-113 and line 152: you wrote "the chamber is filled with hexafluoroethane (an electrically insulating gas)" and "For tests in the vacuum chamber, the pressure is kept to $<2 \cdot 10^{-7}$ Torr", respectively. Have you tried to perform measurements in a non-controlled chamber, i.e. in a realistic environment? This is obviously connected to my previous point.

5. Lines 136-137: you wrote "The rods are cut to the standard longitudinal extension mode Y_{360} orientation and are diamond core drilled to diameters between 9.5 – 12.7 mm". What about cost feasibility of the proposed solution? And what about diameter, which is 8 mm in the tested device (not 9.5 mm)?

6. Line 149: you wrote "field shaping toroid". Please, comment more on how the toroids shape the field.

Other minor comments:

7. What is, according to your calculations, the maximum power level that can be transmitted by your piezoelectric antenna?

8. Have you thought of its practical integration in a transmitting system to make it really "portable"?

Response to Reviewers

Reviewer #1 (Remarks to the Author):

The authors demonstrate an ultra-low loss lithium niobate piezoelectric electric dipole driven at acoustic resonance that radiates with higher efficiency compared to previous devices. Very Low Frequency (VLF) communication systems (3 kHz-50 kHz) are a hot topic in current literature. The paper is well presented and discussed. A nice piece of work. Deserves publication as it is.

We thank you for your comments.

Reviewer #2 (Remarks to the Author):

Dear authors,

I have read very carefully your paper about a high-Q piezoelectric resonator as a portable VLF transmitter. I have found very impressive the amount of work spent in designing, fabricating, testing and modelling the proposed device. I think that that your antenna demonstrates the possibility of achieving better performance with electrically-small antennas (ESAs) with respect to the actual state-of-the-art and, consequently, could have future practical applications for VLF transmitters. Furthermore, the analysis of the performance is, in my opinion, carried out correctly and provides enough details for a reproducibility at laboratory level. References are appropriate, too, and embrace the most recent results in the field of ESAs.

We thank you for your comments.

However, I have some observations that maybe could help enhance the paper:

1. Lines 41-42: you wrote "Large, external impedance-matching elements are superfluous when the device is self-resonant". I do not agree completely: in principle what you stated is true, as an ESA

exhibits a low radiation resistance (sometimes close to zero) and an impedance transformer could be sufficient, but in general impedance matching depends also on the application, for example the self-resonating dipole could be connected to a highly-reactive impedance chip.

Agreed that the statement was too broad. We have softened to, "...much less than the electromagnetic wavelength, potentially removing the need for large, external impedance-matching elements."

2. Lines 75-76: you wrote "The velocity measurement is non-intrusive to the resonator and directly correlates to the dipole moment". I think that this aspect should be explained more in detail.

The correlation can be derived from first-principles, but is outside the scope of the paper. To better describe our intent and how we actually used the measurement, the text has been changed to: "The velocity measurement is non-intrusive to the resonator operation and can be used to calculate the dipole moment. Multiphysics simulations show a one to one correlation of velocity with dipole moment near resonance. We use this attribute to more easily characterize transmitter behavior in a controlled laboratory setting."

3. Line 78: you talk about near field: what is the distance used for your measurements (I think something like a couple of meters)? As an electrical/electronic engineer, I think that it could be even more useful to the scientific community to provide, if possible, some measurements at a far greater distance (not at far field of course, which is not practicable at this frequency).

[Redacted]

4. Lines 112-113 and line 152: you wrote "the chamber is filled with hexafluoroethane (an electrically insulating gas)" and "For tests in the vacuum chamber, the pressure is kept to $<2 \cdot 10^{-7}$ Torr", respectively. Have you tried to perform measurements in a non-controlled chamber, i.e. in a realistic environment? This is obviously connected to my previous point.

We envision one way to utilize this concept is within a portable dielectric chamber. However, operation in ambient air is also possible. In fact, all of our tests at range were with the piezoelectric device in open air. We've added a photograph of the portable system to Fig. 15. Of course, this is a prototype proof of concept, so we expect further reduction in size as the concept matures.

We've now clarified the text which previously incorrectly implied a controlled environment for outdoor tests.

5. Lines 136-137: you wrote "The rods are cut to the standard longitudinal extension mode Y<360 orientation and are diamond core drilled to diameters between 9.5 – 12.7 mm". What about cost feasibility of the proposed solution? And what about diameter, which is 8 mm in the tested device (not 9.5 mm)?

Our goal for this study is not to cost-engineer the device nor target a specific application. However, the LN cut that we utilize is a very common and was not special for our application.

The diameter for the device used in this study is 16.2 mm. You are correct that our quoted core-drill range is not consistent with this diameter. In fact, for the particular LN piece used in this study, we did not utilize a core drill, we machined a rod from cuboid pieces. We have corrected the text in the methods section. The previous text is consistent with devices we are using in other similar studies. In addition, we corrected the typo in the body of the paper which incorrectly identified the radius as diameter.

6. Line 149: you wrote "field shaping toroid". Please, comment more on how the toroids shape the field.

The following text has been added: "...to a field shaping toroid. Using this common high-voltage design technique, the toroids are at or near the potential of the LN rod corners and therefore spatially distribute the equipotential lines. This decreases the peak surface electric field on the LN for a given dipole moment, thereby increasing the achievable dipole moment prior to high voltage flashover."

Other minor comments:

7. What is, according to your calculations, the maximum power level that can be transmitted by your piezoelectric antenna?

As described in the text, the present version has achieved an electric dipole moment of ~ 7 mA-m, rms. This is not a fundamental limit and is dependent upon factors such as LN fracture and volume. The power radiated scales as the dipole moment squared. However, as a convention we prefer to use the quantity *dipole moment* rather than *power* to describe the behavior.

8. Have you thought of its practical integration in a transmitting system to make it really "portable"?

We've now included a photograph of the prototype transmitter in Fig. 15. This device is hand portable.

REVIEWERS' COMMENTS:

Reviewer #2 (Remarks to the Author):

Dear authors,

I am very satisfied about your point-to-point responses. The paper can be published as it is. Nice work!